# Bending strain engineering in quantum spin hall system for controlling spin currents

Bing Huang[1], Kyung-Hwan Jin[2], Bin Cui[2], Feng Zhai[3], Jiawei Mei[1,2] & Feng Liu[2,4]

Quantum spin Hall system can exhibit exotic spin transport phenomena, mediated by its topological edge states. Here the concept of bending strain engineering to tune the spin transport properties of a quantum spin Hall system is demonstrated. We show that bending strain can be used to control the spin orientation of counter-propagating edge states of a quantum spin system to generate a non-zero spin current. This physics mechanism can be applied to effectively tune the spin current and pure spin current decoupled from charge current in a quantum spin Hall system by control of its bending curvature. Furthermore, the curved quantum spin Hall system can be achieved by the concept of topological nanomechanical architecture in a controllable way, as demonstrated by the material example of Bi/Cl/Si(111) nanofilm. This concept of bending strain engineering of spins via topological nanomechanical architecture affords a promising route towards the realization of topological nano-mechanospintronics.

[1] Beijing Computational Science Research Center, Beijing 100193, China. [2] Department of Materials Science and Engineering, University of Utah, Salt Lake City, Utah 84112, USA. [3] Department of Physics, Zhejiang Normal University, Jinhua 321004, China. [4] Collaborative Innovation Center of Quantum Matter, Beijing 100084, China. Correspondence and requests for materials should be addressed to B.H. (email: bing.huang@csrc.ac.cn) or to F.L. (email: fliu@eng.utah.edu).

A long-standing interest in spintronics is generating and transporting spin current (SC) in condensed matter systems. In the past decades, significant process has been made towards realization of highly polarized SC with ferromagnetic materials[1,2], in which SC is strongly coupled with charge current (CC). The discovery of pure spin current (PSC), for example, spin Hall current, that is decoupled from CC[3,4] has opened up exciting opportunities for spin transport, because it is expected that the transport of PSC has much smaller energy dissipation compared with that of conventional SC generated by ferromagnetic materials. Quantum spin Hall (QSH) system can exhibit exotic spin transport properties[5,6], especially, a transverse edge PSC of QSH effect can be generated under a four-terminal device setting. For a conventional flat QSH insulator, there are two basic properties, time reversal symmetry (TRS) and spin conservation, which are of special interest. TRS renders the edge states of a QSH insulator topologically protected to transport robust SC without elastic back-scattering from non-magnetic impurities. However, spin conservation mandates that there is no net SC under a two-terminal device setting in a QSH system[5]. Although discovering new mechanism to control the SC and/or transverse PSC in a QSH system is of great importance for spintronics, its development is still at its infancy.

Strain engineering has been developed as a well-established approach to enhance the performance of electronic devices, such as Si transistors[7], by tuning band structure and carrier mobility of semiconductors[8,9]. Recently, strain engineering has been extended to create interesting physical phenomena in 2D materials[10,11], for example, pseudo-magnetic fields[12–15] and superconductivity[16] in graphene. Moreover, strain engineering has also been exploited in materials fabrication through strain induced self-assembly of nanostructures in heteroepitaxial growth of thin films[17–19] and most recently through strain partitioned nanomembranes and nanomechanical architecture[20].

In the same spirit of conventional strain engineering of electronic properties, the strain engineering of topological properties has been recognized[6,21], because strain changes the bulk band gap of TIs inducing topological phase transitions. Usually, the form of strain considered is tensional strain via lattice expansion/compression. In this article, we explore a form of bending strain engineering to tune the spin transport of QSH edge states by curvature effect. We demonstrate that for a QSH system under bending strain, curvature preserves its TRS but mitigates spin conservation, so that a spin torque occurs to generate a non-zero SC under a two-terminal device setting, which can make this system working as a topological half-metal under a bias. This idea can further be applied to control the magnitude of transverse PSC of a QSH system by control of its bending curvature, which has not been achieved in a QSH system before. In terms of material design, we suggest a possible approach to grow the self-bending QSH systems via the concept of 'topological nanomechanical architecture', as demonstrated by the material example of Bi/Cl/Si(111) nanofilm, which may pave the way for the realization and study of topological nano-mechanospintronics.

## Results

**Model of curved quantum spin Hall systems.** We start from a curved QSH system on a hexagonal lattice, as shown in Fig. 1a. We define a center angle between the left and right edge of a curved ribbon, $\theta_e$, to represent the magnitude of bending curvature. Following Kane and Mele[5], a QSH Hamiltonian contains two minimal terms, $H = H_0 + H_{so}$. Assuming a sufficiently large spin-orbit coupling (SOC), the gap is

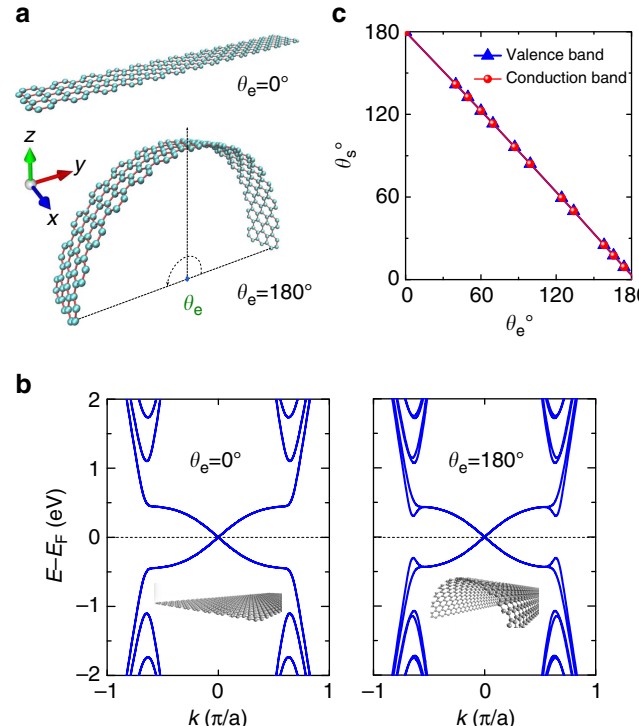

**Figure 1 | Curvature dependent electronic structures of QSH ribbons.**
(**a**) Flat and curved structures of a zigzag-edge ribbon with hexagonal lattice. The $\theta_e$, defined as the center angle between the two edges of the curved structure, effectively represents the bending curvature. The coordinate axes are also indicated. The periodic direction of ribbon is along $x$ direction. (**b**) Band structures of flat ($\theta_e = 0°$) and curved ($\theta_e = 180°$) zigzag-edge ribbons with 40 atoms per unitcell. (**c**) Spin angles $\theta_s$ (for both conduction and valence bands) of two double-degenerated edge states as a function of $\theta_e$.

insensitive to the changes in hopping or a small staggering potential, then the bending will not qualitatively change the $H_0$ term. Thus, our central attention will be the SOC term. Especially, bending changes the directions of orbital angular momenta, which in turn changes the spin directions subject to the spin-momentum locking property. As we will show below, mechanical bending can generate non-zero spin conductance in a curved QSH system under a two-terminal device setting. All the features of curved QSH systems are intrinsic, which will have profound effects on spin transport properties, independent of specific QSH materials considered.

Concretely, we study a simplified ($p_x$, $p_x$) four-band model Hamiltonian in a hexagonal lattice[22,23] (Fig. 1a, upper panel), we have

$$H_0 = \sum_{n,i} \varepsilon_{ni} c_{ni}^\dagger c_{ni} - \sum_{\langle mj,ni \rangle} t_{mj,ni} \left( c_{mj}^\dagger c_{ni} + h.c. \right), \quad (1)$$

$$H_{so} = i\lambda_{so} \sum_{\langle\langle ni,nj \rangle\rangle} c_{ni}^\dagger \boldsymbol{\sigma} \cdot \left( \boldsymbol{e}_{ni} \times \boldsymbol{e}_{nj} \right) c_{nj}. \quad (2)$$

In $H_0$, $c_{ni}^\dagger = (c_{ni\uparrow}^\dagger, c_{ni\downarrow}^\dagger)$ are electron creation operators on atom $n$ with orbital $i \in \{p_x, p_y\}$; $\varepsilon_{ni}$ and $t_{mj,ni}$ are electronic on-site energies and hopping integrals, respectively. $t_{mj,ni} = A_{mj,ni}(pp\sigma) + (\boldsymbol{e}_{mj} \cdot \boldsymbol{e}_{ni} - A_{mj,ni})(pp\pi)$, where $A_{mj,ni} = (\boldsymbol{e}_{mj} \cdot \boldsymbol{e}_{mn})(\boldsymbol{e}_{ni} \cdot \boldsymbol{e}_{mn})$. $\boldsymbol{e}_{ni}$ represents the unit vector along the orbital $i$ of atom $n$, and $\boldsymbol{e}_{mn}$ is the unit vector directed from site $m$ to $n$. $(pp\sigma)$ and $(pp\pi)$ are Slater–Koster integrals[24]. Thus, the value of $t_{mj,ni}$ depends on the

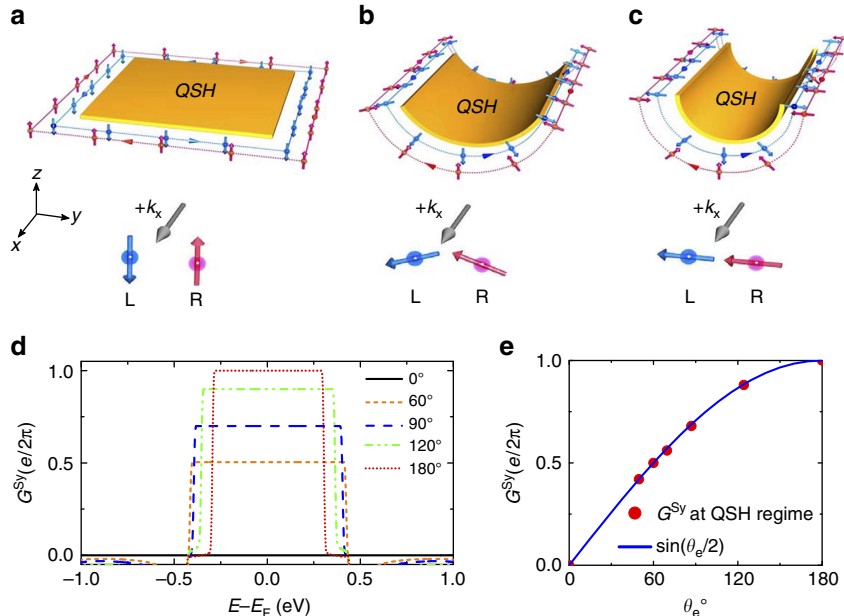

**Figure 2 | Curvature dependent spin conductances of QSH ribbons.** (**a**–**c**) Schematic diagrams of spin current and charge current flowing along the edges as the $\theta_e$ increases from 0° (**a**) to 180° (**c**). A pair of edge states counter propagate along all four edges subject to TRS. The spins rotate adiabatically along the curved edges. The highlight of spin directions at the two opposite edges under the same charge current flow direction $+k_x$ is shown in the bottom of **a**–**c**. (**d**) Calculated spin conductance $G^{S_y}$ $(G^{S_x} = G^{S_z} = 0)$ for the QSH ribbons with different $\theta_e$ in a two-terminal device setting. (**e**) The values of $G^{S_y}$ in the QSH regime (plateau region) in **d** as a function of $\theta_e$, which can be perfectly described by the equation of $G^{S_y} = \sin(\frac{\theta_e}{2}) \cdot \frac{e}{2\pi}$.

relative position and directional cosines between $mj$ and $ni$ orbitals at a given $\theta_e$. When $\theta_e$ is changed, the contribution (projection) of $(pp\sigma)$ and $(pp\pi)$ to $t_{mj,ni}$ changes. In our calculations, $\varepsilon_{ni}$ are set to zero and only the nearest neighbour (NN) hoppings are taken into account. In $H_{so}$, $\lambda_{so}$ is a constant, defining the SOC strength, and $\boldsymbol{\sigma}$ is the Pauli vector. In the flat QSH system, $\theta_e = 0°$, $\boldsymbol{e}_{ni} \times \boldsymbol{e}_{nj} = \pm \boldsymbol{e}_z$, and spins lie strictly along the $z$ direction. When $\theta_e \neq 0°$, upon bending, all the physical observables, for example, $p_{x,y}$ orbitals, $\boldsymbol{e}_{ni}$ and spin directions $\boldsymbol{\sigma}$ in equations (1 and 2), are rotated accordingly by an angle of $\phi$ relative to the $x$-axis (see Supplementary Fig. 1 for the definition of $\phi$).

Figure 1b shows the band structures of flat ($\theta_e = 0°$) and curved ($\theta_e = 180°$) QSH ribbons (see Supplementary Fig. 2 for the band structures of other $\theta_e$ cases), calculated using the TB parameters of $(pp\sigma) = 6.38$ eV, $(pp\pi) = -2.66$ eV (refs 24,25) and $\lambda_{so} = 0.9$ eV. The topological Dirac edge states are clearly seen around Fermi level in both cases, which are double-degenerated. For $\theta_e = 0°$, the two forward-propagating edge states along opposite boundary have opposite spin orientations along $\pm z$ axis, and the spin $\boldsymbol{S} = \frac{\hbar}{2}\boldsymbol{\sigma}$ of edge state Pauli matrix can be described by $\sigma_z$ basis. The spin angle $\theta_s$, defined as the angle between the spin vectors of two edge states, is 180° (Fig. 1c) for $\theta_e = 0°$. For $\theta_e \neq 0°$, bending shows little effect on the shape and degeneracy of edge states, but it significantly changes spin orientation, as shown in Fig. 1c. On bending, $\boldsymbol{S}$ can be expressed as a linear combination of $\sigma_y$ and $\sigma_z$, and the net spin direction in the whole system is along the $y$ axis, which can be expressed in $\sigma_y$ basis. The larger the $\theta_e$, the larger the net $\sigma_y$ spin component. Our calculations establish a simple relationship between $\theta_s$ and $\theta_e$ as $\theta_s + \theta_e = 180°$.

**Spin transport in curved quantum spin Hall systems**. Curvature does not remove TRS in curved QSH systems, and spin/charge currents with opposite polarity still propagates in opposite directions along the edges, as shown in Fig. 2a–c (from Fig. 2a–c:

$\theta_e$ increases from 0° to 180°), which is also reflected by the unchanged edge band structures (Fig. 1b). However, curvature mitigates spin conservation; spins are no longer conserved along the edges, for example, they adiabatically rotate on the curved sides of edges, which is expected to modify non-equilibrium spin transport properties in curved QSH systems under a bias. Specifically, edge spin rotation in Fig. 2a–c is achieved by creating an $S_y$ component in addition to $S_z$. At two opposite edges of a QSH ribbon, the $S_z$ components are antiparallel (pointing in the opposite directions at opposite edges) but the $S_y$ components are parallel (pointing in the same directions at opposite edges) to each other along the same direction of charge current. Consequently, a conventional (flat) QSH insulator conducts only CC but not SC under a two-terminal device setting because only the $S_z$ component is present, while a curved QSH insulator can conduct both CC and SC arising from the emergence of $S_y$ component.

Quantitatively, we can calculate the two-terminal charge and spin transmission coefficients of QSH ribbons with different $\theta_e$ using non-equilibrium Green's function formalism in the linear-response regime[26] as

$$T_\alpha(E) = \text{Tr}[\sigma_\alpha \Gamma_L \mathbb{G}^r(E) \Gamma_R \mathbb{G}^a(E)], \qquad (3)$$

where $\Gamma_{L/R} = i\left[\Sigma_{L/R} - \Sigma_{L/R}^\dagger\right]$ indicates the interaction between a central scattering area and left/right lead, whose self-energy is $\Sigma_{L/R}$. $\mathbb{G}^{r/a}(E)$ is the retarded/advanced Green's function, whose definition is $\mathbb{G}^r(E) = [H - (E + i\eta) + \Sigma_L + \Sigma_R]^{-1}$ or $\mathbb{G}^a(E) = (\mathbb{G}^r)^\dagger$. When $\alpha = 0$, $\sigma_\alpha$ represents unit matrix for charge conductance; when $\alpha = x, y, z$, $\sigma_\alpha$ represents the Pauli matrices for spin conductance[27]. Both source/drain and central scattering area are made of the same material to reveal the intrinsic transport properties.

At the QSH regime (plateau region), our calculations show that the charge conductance $G$ is insensitive to curvature and remains at its quantized value, as shown in Supplementary Fig. 3, but the spin conductance $G^{S_y}$ (here $G^{S_x} = G^{S_z} = 0$) becomes

$\theta_e$-dependent and no longer quantized. As shown in Fig. 2d, $G^{S_y}$ gradually increases from 0 ($\theta_e = 0°$) to $\frac{e}{2\pi}$ ($\theta_e = 180°$) with the increasing $\theta_e$ in the QSH regime under a two-terminal device setting, consistent with the increased $S_y$ component (Fig. 1c). The spin conductance $G^S$ can only take the forward-propagating edge modes, whose direction is given by the direction of external bias.

Furthermore, we can make some general arguments to illustrate the effects of $\theta_e$ on the $G^S$ of a QSH system. Actually it is legal for us to implement a local coordinate (unitary) transformation on the curved QSH system

$$H_C \rightarrow \tilde{H} = \mathcal{R}^\dagger H_C \mathcal{R}, \quad \mathcal{R} = \prod_m R_m^d R_m^s, \qquad (4)$$

such that we can transform our curved system $H_C$ into a flat one $\tilde{H}$. Here $R_m^d$ is spin-independent deformations and $R_m^s = e^{i\frac{\sum_i (S_x)_m^i \phi(m)}{\hbar}}$ is the spin rotation for electron operators $c_{mi}$ of all orbitals $i$ on atom $m$. If the Rashba effect $H_R$ is ignored, we can find that the total rotated spin $z$-component $\tilde{S}_z = \mathcal{R}^\dagger S_z \mathcal{R}$ is conserved, $[\tilde{S}_z, \tilde{H}_C] = 0$, and the spin Chern number $\tilde{C}_s$ in the flat system is well defined[28]. The spin conductances at both left and right edges have only non-zero $z$-component $g_{L,R}^{\tilde{S}_z} = \frac{\tilde{C}_s}{2} \cdot \frac{e}{2\pi}$. These results are well established for a flat QSH system. If we go back to the original curved reference frame, we have the edge spin conductance, $g_{L,R}^{S_y} = \sin(\phi_{L,R}) \frac{\tilde{C}_s}{2} \cdot \frac{e}{2\pi}$, $g_{L,R}^{S_z} = \cos(\phi_{L,R}) \frac{\tilde{C}_s}{2} \cdot \frac{e}{2\pi}$ (see details in Supplementary Note 1). By definition, the net spin conductance is given as $G^{S_y} = \frac{g_L^{S_y} - g_R^{S_y}}{2}$, $G^{S_z} = \frac{g_L^{S_z} - g_R^{S_z}}{2}$. Therefore, in the curved system, $\phi(L) = -\phi(R) = \frac{\theta_e}{2}$, $\tilde{C}_s = 2$, then spin conductance is

$$G^{S_y} = sin\left(\frac{\theta_e}{2}\right) \cdot \frac{e}{2\pi}, \quad G^{S_z} = 0. \qquad (5)$$

This conclusion agrees well with our calculations, as shown in Fig. 2e. We want to emphasize that all these features of curved QSH systems are intrinsic, having profound effects on spin transport properties independent of specific QSH materials considered.

**Tunable spin currents in curved quantum spin Hall systems.** On the basis of the same physical mechanism, curvature can also modify the transverse PSC of QSH systems, because this PSC is only contributed by the $S_z$ components which decreases with the increased $S_y$ components, and the quantization of the spin Hall conductance in a QSH system is only guaranteed when $S_z$ is conserved[29].

More generally, we provide a comparison between the charge and spin transport properties of curved QSH devices and those of conventional (flat) QSH devices in both two- and four-terminal device settings within the Landauer–Büttiker[30] framework, as shown in Fig. 3. In terms of transport, with two terminals, the curved QSH (Fig. 3b,c, upper panel) device conducts both CC and SC (0 to $\left(\frac{e}{2\pi}\right)V$), which is significantly different from the flat QSH device that conducts only CC (Fig. 3a, upper panel). A curved QSH device can effectively work as a topological half-metal for spin injection, that is, it transports topologically protected completely spin-polarized charge current, and the density of SC can be tuned by the curvature. With four terminals, the flat QSH device conducts a longitudinal CC ($I_1$) and a transverse PSC ($I_t^s$) (Fig. 3a, lower panel), while the curved QSH device with $0 < \theta_e < 180°$ (Fig. 3b, lower panel) conducts both longitudinal CC $I_1$ and SC $I_l^s$ (contributed by $S_y$ component), as well as a transverse PSC $I_t^s$ (contributed by $S_z$ component). Interestingly, $I_t^s$ ($I_l^s$) continues to decrease (increase) with increasing $\theta_e$ subject to

the conservation of total spin, $S = S_y + S_z$, and finally $I_t^s$ vanishes at $\theta_e = 180°$ (Fig. 3c, lower panel).

In terms of robustness against elastic back-scattering from non-magnetic impurities, the curved and flat QSH devices are the same, as they are protected by TRS. In terms of conductance quantization, charge conductance is integer-quantized in unit of $\frac{e^2}{h}$ in both flat and curved QSH devices, as shown in Supplementary Fig. 3. However, spin conductance is only integer-quantized in unit of $\frac{e}{4\pi}$ in the flat but not in the curved QSH device, hence in the latter the spin conductance, arising from the $S_y$ components, is not conserved for different $\theta_e$ (Fig. 2e). Moreover, the curved QSH systems can also exhibit some similar transport properties to quantum anomalous Hall systems, as shown in Supplementary Fig. 4. Therefore, curvature, induced by bending strain, can be employed to dramatically tune the topological SC and transverse PSC in the a curved QSH system for various spintronics applications.

**Rashba and disorder effects.** It is interesting to consider the Rashba and other disorder effects on the electronic and transport properties of curved QSH systems. First, we have considered a simplified NN hopping Rashba spin-orbital term $H_R$ (refs 5,31) in which the difference of symmetry between $p_x$ and $p_y$ orbitals are neglected:

$$H_R = i\lambda_R \sum_{\langle mj, ni \rangle} c_{mj}^\dagger (\boldsymbol{\sigma} \times \boldsymbol{e}_{mn})_z c_{ni}. \qquad (6)$$

The strength of $H_R$ is determined by $\lambda_R$. Thus, the Hamiltonian becomes $H = H_0 + H_{so} + H_R$. As shown in Supplementary Fig. 5, when $\lambda_R$ is increased from 0 to 0.9 eV (the value of $\lambda_{so}$), the Rashba effect can significantly reduce the original SOC band gap, but the topological properties are unchanged as long as the SOC band gap is not closed. Importantly, the Rashba effect will not change the spin orientation of edge states and the curvature can still be applied to generate a significant non-zero spin conductance in a QSH system, as demonstrated in Supplementary Fig. 6.

Second, we have considered the effects due to random on-site energies $\varepsilon_{ni}$, that is, by randomly changing the $\varepsilon_{ni}$ of all the atoms in $H_0$ term in equation (1), which can simulate the effects from substrate or impurity or by many-body interactions through self-energy corrections. The variations of the $\varepsilon_{ni}$ from the initial values up to $1.0\lambda_{so}$ are considered, as shown in Supplementary Fig. 7. The random $\varepsilon_{ni}$ can effectively lift the degeneracy of edge states, but it will not affect the spin rotations of edge states. As shown in Supplementary Fig. 8, random $\varepsilon_{ni}$ effects will not change the overall shape of charge and spin conductance spectrums or alter our main conclusions.

Third, we have considered the random-atomic-position (RAP) effect, which can account for the electron–phonon interactions or thermal effect. We assume all the atoms are displaced from their equilibrium positions in any given direction by a maximum distance of 0.125 Å. This will effectively change the hopping term $t_{mj,ni}$ in $H_0$ in equation (1), even if the changes in $(pp\sigma)$ and $(pp\pi)$ are negligible. Similar to the random $\varepsilon_{ni}$ effect, the RAP effect also lifts the degeneracy of edge states, as shown in Supplementary Fig. 9. However, it will not affect the transport properties of QSH ribbons, as shown in Supplementary Fig. 10.

Finally, it is also important to note that inelastic back scattering can still occur in the presence of many-body interactions in a QSH system, which may induce a finite conductivity[32]. However, the recent experiment measurements in InAs/GaSb bilayer[33] system indicate that even the helical edge modes are in a strongly interacting regime, the quantized conductance plateaus can still survive in a broad regime.

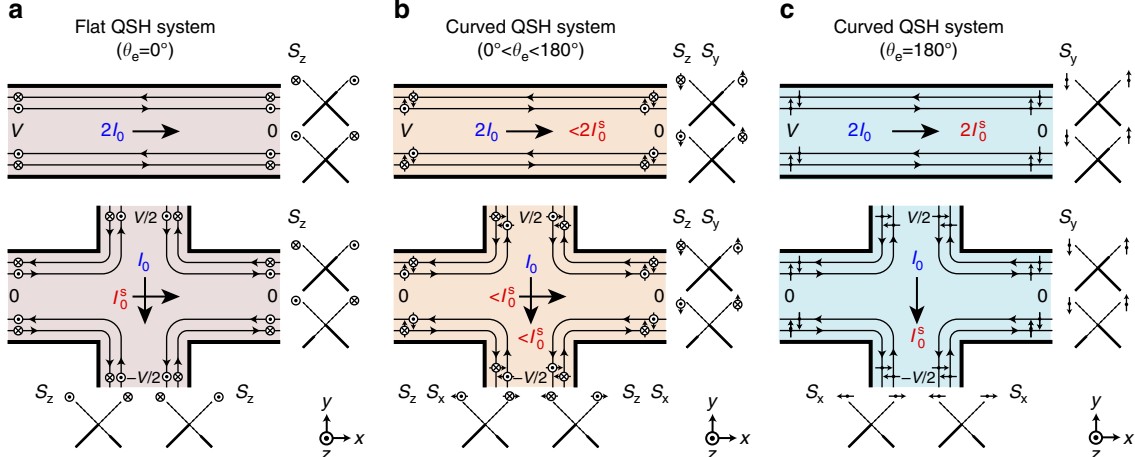

**Figure 3 | Curvature dependent spin currents in QSH devices.** Comparison of two-terminal and four-terminal measurement geometries for a (**a**) flat QSH system ($\theta_e = 0°$), (**b**) curved QSH system ($0 < \theta_e < 180°$) and (**c**) curved QSH system ($\theta_e = 180°$). The arrows indicate the charge current $I$ and spin current $I^s$ and their flow directions. The unit of $I$ and $I^s$ are $I_0 = (e^2/h)V$ and $I_0^s = (e/4\pi)V$, respectively. The diagrams to the right and bottom indicate population of the edge states.

**Concept of topological nanomechanical architecture.** A practical idea to realize bent QSH systems is nanomechanical architecture of strained nanofilms, which has been proven a powerful method to fabricate nanomembranes, nanotubes, partial or half nanotubes, and nanocoils[17,18,34]. The general process of nanomechanical architecture proceeds with growth of strained nanofilms on a sacrificial substrate followed by patterning and release (through removal of the sacrificial substrate) of the nanofilms, which will roll-up into different tubular shapes as pre-designed by strain engineering (see Supplementary Movie 1 for this concept). Suppose one can apply the same process to a QSH nanofilm, then strain engineering of topological boundary states is realized to tune the edge spin orientations in a controllable manner.

Furthermore, it is a parallel process that can facilitate mass production of identical partial cylindrical QSH arrays[35], which will function ideally as a robust spin injector device with high spin current density, as demonstrated in Supplementary Fig. 11, while spin polarization can be switched by changing the bias direction. Compared with the traditional magnetic materials, for example, ferromagnetic metals, the QSH system based spin injectors are topologically protected, robust against structural distortion or impurity scattering; the helical Dirac edge states support also ultra-fast SC transport.

**Self-bending behaviours of Bi/Cl/Si(111) films.** To demonstrate the feasibility of the above concept, we have further performed first-principles calculations to study the evolution of topological edges states of a QSH Bi/Cl/Si(111) nanofilm under self-bending driven by the nanomechanical architecture process. It has been predicted that a surface based QSH state forms in a hexagonal Bi overlayer deposited in the halogenated Si(111) surface, that is, Bi/Cl/Si(111)[36]. If one grows a ultrathin Si(111) film on a sacrificial SiO$_2$ substrate before Cl adsorption and Bi deposition, then the resulting Bi/Cl/Si(111) nanofilm is readily subject to the nanomechanical architectural process, sell-rolling into a tubular shape (including a partial cylinder) on releasing from the underlying SiO$_2$ substrate.

The Si(111) surface functionalized with one-third monolayer (ML) of Cl exhibits a $\sqrt{3} \times \sqrt{3}$ reconstruction[37,38]. When 1 ML Bi is deposited on the Cl/Si(111) surface, the most stable structure of Bi atoms adopts a hexagonal Bi lattice (Fig. 4a)[36]. The Bi lattice has an in-plane lattice constant of 3.87 Å, $\sim$20% larger than that

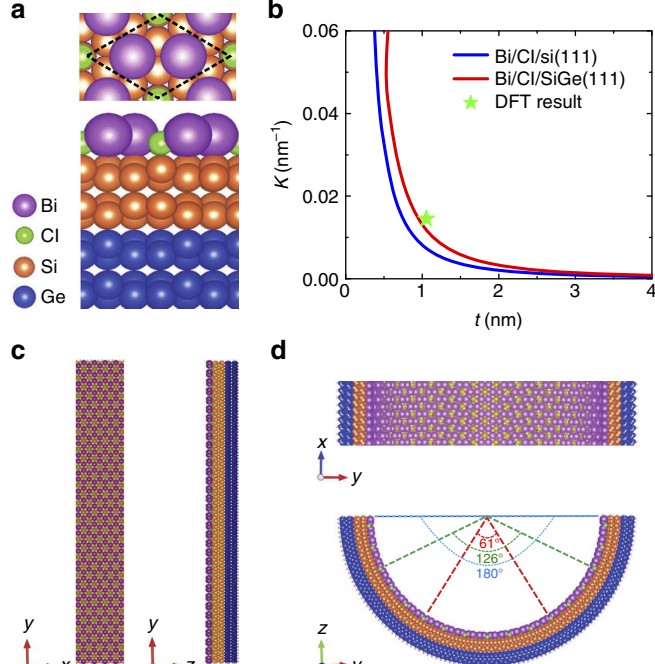

**Figure 4 | Self-bending behaviour of a QSH film.** (**a**) Top and side views of a Bi/Cl/SiGe(111) surface. The dashed black lines mark the unit cell. (**b**) The calculated self-bending curvatures of Bi/Cl/Si(111) and Bi/Cl/SiGe(111) nanofilm as a function of film thickness based on Stoney and Timoshenko formulas, respectively. The DFT simulated bending curvature of a Bi/Cl/SiGe(111) of a 1.05 nm thickness is also shown (a star). (**c**) Top and side views of a flat Bi/Cl/SiGe(111) surface with two atomic layers Si and Ge each. The ribbon edge termination is along the zigzag-edge direction of Bi lattice (edges are passivated by H atoms). (**d**) Self-bent structure shown in **c** obtained from first-principles total energy relaxation. Three different angles in **d** represent three ribbon widths (along the y direction) with different edge bending angles $\theta_e$.

of free-standing Bi layer. This gives rise to a large tensile surface stress of 0.12 eV Å$^{-1}$ in the top surface of Bi/Cl/Si(111), obtained from first-principles calculations. On the other hand, the bottom surface of Bi/Cl/Si(111), which might be bare (or H-passivated) during the release process from the underneath substrate, has a

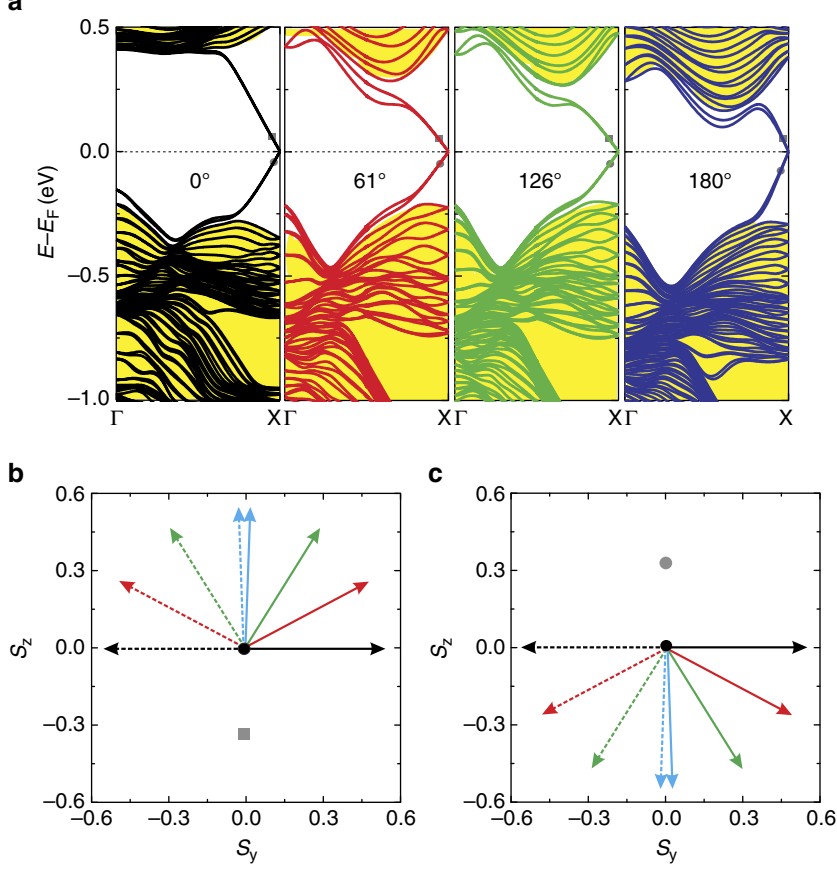

**Figure 5 | Electronic structures of Bi/Cl/SiGe(111) ribbons. (a)** First-principles calculated band structures of four Bi/Cl/SiGe(111) ribbons at different bending angle $\theta_e$, 0°, 61°, 126° and 180°, respectively. The bulk bands are marked in yellow region. **(b)** Spin rotations of conduction band edge states at a $k$ momentum slightly off Dirac point (X), as marked in **a**. **(c)** Same as **b** but for valence band edge states.

much smaller surface stress of $-0.04$ (or $\sim 0.00\,\text{eV}\,\text{Å}^{-1}$). Therefore, there can exist a stress imbalance between the top and bottom surface of Bi/Cl/Si(111) nanofilm, which provides a driving force for self-bending. Using the surface stress difference $\Delta\sigma$ as input, we can estimate the bending curvature $\kappa$ of Bi/Cl/Si(111) nanofilm as a function of film thickness $t$ using Stoney formula[39] $\kappa = (6\Delta\sigma)/(Ct^2)$, as shown in Fig. 4b, where $C = E/(1-v^2)$ is a constant related to Young's modulus $E$ and Poisson ratio $v$ of Si.

As an important strategy in nanomechanical architecture, besides changing film thickness, another effective way to control the bending curvature of the rolled-up tubular structure is to grow lattice-mismatched multilayer film to partition the amount of misfit strain and tune the driving force for bending. Specifically, SiGe film is often used for this purpose, as Ge lattice is $\sim 4.2\%$ larger than Si lattice and the growth SiGe film is a well-established technique. To verify this idea, we have taken the bilayer system of Bi/Cl/SiGe(111) film (two atomic layers of Si and Ge each) as an example, and the calculated total imbalanced 'surface' stress in this system is $0.21\,\text{eV}\,\text{Å}^{-1}$, about two times larger than that of Bi/Cl/Si(111). We can estimate the bending curvature of Bi/Cl/SiGe(111) nanofilm as a function of total film thickness $t$ from Timoshenko formula[40] $\kappa = (6\Delta\sigma/E_s t^2)\gamma$ and $\gamma = \frac{(1+\beta)^3}{1+4\alpha\beta+6\alpha\beta^2+4\alpha\beta^3+\alpha^2\beta^4}$, where $\alpha = E_f/E_s$, $E_f$ and $E_s$ are Young's modulus of Si and Ge, respectively, $\beta = t_f/t_s$ is the ratio of Si thickness $t_f$ and Ge thickness $t_s$, and $t = t_f + t_s$. The result is shown in Fig. 4b, confirming a larger bending curvature than Bi/Cl/Si (111) system.

Next, we use first-principles calculations to directly simulate the self-bending curvature of a nanoribbon of finite width made of Bi/Cl/SiGe(111) nanofilm, as shown in Fig. 4c,d. For comparison, we again choose two atomic layers of Si and Ge. The edges are along the zigzag edge direction of Bi lattice and passivated with H atoms to remove dangling bonds. The calculated self-bending curvature of Bi/Cl/SiGe(111) is $0.0136\,\text{nm}^{-1}$, which agrees quite well with the estimation from Timoshenko formula, that is, $0.0119\,\text{nm}^{-1}$.

**Electronic properties of curved Bi/Cl/SiGe(111) films.** After the self-bending curvature of Bi/Cl/SiGe(111) is determined, three different ribbon widths are used to simulate three different $\theta_e$, which are 61°, 126° and 180°, as indicated, respectively, in Fig. 4d, for topological edge state calculations. As Bi $p_z$ orbitals are passivated by the top Si atoms on the substrate, the remaining Bi $p_x + p_y$ orbitals realize a QSH phase, which can be described by a four-band TB model Hamiltonian of equations (1 and 2). The calculated band structures for these three $\theta_e$ cases, along with the case of $\theta_e = 0°$, is shown in Fig. 5a, where the existence of linearly dispersive Dirac bands crossing the Fermi level indicates a nontrivial band topology. The Dirac edge states persist with bending, as expected from their topological origin to be robust against structural deformation. After bending, the degeneracy of the two edge states are slightly lifted when the energy moves away from the Fermi level because of the broken symmetry.

Figure 5b,c shows the evolution of the spin direction of the conduction and valence edge states slightly off the Dirac point.

For $\theta_e = 0°$, the spins are aligned normal to the ribbon plane ($z$-axis) and the two edge spins are orientated antiparallel ($\theta_s = 180°$) with each other. On bending, as shown in Fig. 5b, for conduction band edge states the spins rotate counterclockwise (clockwise) for the left (right) states as $\theta_e$ increases. When $\theta_e$ reaches $\sim 180°$, spins are rotated into almost parallel along the $y$-axis at both edges ($\theta_s = 4°$). Similar behaviour is found for valence band edge states, as shown in Fig. 5c. Thus, our first-principles calculations of a real QSH material not only can confirm the concept proposed, but also suggest a promising way to realize topological spintronics materials by nanomechanical architecture.

## Discussion

We have theoretically proposed a concept of bending strain engineering of spin transport in QSH systems, which is generally applicable to all QSH materials and especially suited for surface or interface-based QSH states on or inside a thinfilm. It affords a promising route towards realization of robust QSH-based spin injectors with 100% spin polarization. A curved QSH system may be potentially realized by subjecting a QSH nanofilm to nanomechanical architecture process. Our finding opens an interesting avenue to topological nano-mechanospintronics, enabling generation and transport of spin current by mechanical bending of a QSH system. It significantly advances our fundamental knowledge of spin transport properties, as well as broadens the scope of nanotechnology into topological materials and devices and vice versa.

## Methods

**First-principles calculations.** First-principles calculations based on the density functional theory were performed within the generalized gradient approximation of PBE form for the exchange-correlation of electrons as implemented in the VASP Package[41]. The projected-augmented-wave method was used to describe the atomic potentials. The SOC was included at the second variational step using the scalar-relativistic eigen-functions as a basis. A cutoff energy of 450 eV was used for the expansion of wave functions and potentials in the plane-wave basis. Sufficient k-point meshes were used for sampling the Brillouin zone. The atomic structures of all the calculated systems were fully relaxed until the Helmann-Feynman forces were $< 0.02$ eV Å$^{-1}$. To simulate the nanoribbon structures in the plane-wave basis, we employed the supercell method. Both the edge-to-edge and layer-to-layer distances between adjacent ribbons are set $> 20$ Å, to eliminate artificial interactions between neighbouring cells.

**Data availability.** The data that support the findings of this study are available from the first author and corresponding author on request.

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

## Acknowledgements

B.H., K.H.J., and F.L. acknowledge the support from US-DOE (Grant No. DE-FG02-04ER46148). B.H. also acknowledges the support from NSFC (Grant No. 11574024) and NSAF U1530401. Computations were performed at DOE-NERSC and CHPC of University of Utah and Tianhe2-JK at CSRC.

## Author contributions

F.L. and B.H. directed the project. B.H., K.J., B.C., F.Z. and J.M. calculated and analysed the results. B.H. and F.L. wrote the manuscript. All authors discussed the results and commented on the manuscript.

## Additional information

**Competing interests:** The authors declare no competing financial interests.

