## [Peer Review File · Nature Communications]

Reviewers' comments:

Reviewer #1 (Remarks to the Author):

Summary

In this work, Huang & co-workers present a strain-engineering approach to spin current manipulation in quantum spin Hall (QSH) insulators. The proposed adiabatic manipulation of counter-propagating spins along the edge of a curved QSH – their main result, nicely captured in Fig. 2 (a) – is novel to my best knowledge, and is also very interesting from a theoretical standpoint.

There are a few concerns to keep in mind, notwithstanding: English language issues, mathematical imprecisions, and poor justification for neglecting local curvature effects in their minimal model [Eqs. (1)-(2)]; see below for details. However, I am confident that these issues can be overcome in a revised version, in which case I would be happy to recommend their work for publication in the prestigious Nature Communications venue.

Report

I. Introduction

The authors seem to use “spin current (SC)” with two different meanings. For example, in the first paragraph (Introduction) one can read:

“a QSH insulator [...] transport robust SC without back scattering”
and then

“spin conservation mandates that there is no net SC in a QSH system”

which sounds like a contradiction in terms. The authors need to clarify the distinction between pure spin currents (carrying no charge flow, e.g., as generated by spin Hall effect) and fully/partially-polarised spin currents (carrying carry charge flow, e.g., as generated in a half-metal). A simple fix would be referring to the former as pure spin currents (PSC) and to the latter as spin-polarised currents or simply “spin currents” (SC).

Finally, I note that “spin Hall current” usually means a pure spin transverse (Hall) current generated by spin Hall effect, which in the context of quantum spin Hall effect can be misleading since the spin Hall conductivity (in the bulk) is vanishing.

II. Results

* Section “Model of curved QSH systems”. The text is currently somewhat confusing, requiring some improvements and possibly extra references. For instance, the authors state that term H_0 defines the band structure. This is incorrect: the band structure is defined by all the terms in the Hamiltonian. They also say that bending induces small changes to H_0 , particularly to hopping parameters: can they explain why? Wouldn't local curvature drastically affect the hopping terms, particularly those related to spin-orbit coupling? (Possible useful is the comprehensive work exploring curvature effects in graphene by Huertas-Hernando et al, PRB 74, 155426 2006.) The situation with the px-py-orbital graphene model explored by the authors is different from real pz-type graphene, but I would imagine that that bending would still couple σ (px-py-derived) and π (p_z -derived) bands, boosting Rashba-type SOC, as in pz-derived graphene. In fact, some of the approximations in Ref. 21 (where the minimal model employed by the authors is derived) could become inaccurate upon bending [specifically, I am thinking about the spin flip terms neglected when deriving Eq. (7) from (6) in Ref. 21]. In a revised version, the authors would need to clarify these points so as to fully justify their choice of model, and to provide enough context to the broad readership of Nature.

Other issues with this section.

- The equation (2) is incorrect: creation/ annihilation operators are missing. Moreover, the authors need to define what type of orbitals goes in the index "n" – that this can be inferred from the text is no excuse to abandon rigor.

- Nano-ribbon geometry details (in Fig. 1) are missing: what type of edges, what is the direction "k" in panels b) and c)? These could be inferred from the schematic a) but still would like to see subscripts and other labels correctly displayed. 40 unit cells in what direction?

- TB parameters need to be explained (as for their meaning) in main text or in the supplemental material. For example, how does $V_{pp\sigma}$ relate to their microscopic parameters in eq. 2?

- Finally, can the authors make the nice panel (a) in Fig. 2 larger to help visualising the spin tilting along the edges?

* Section "Spin transport in curved QSH systems"

The phrase

"Spin transport in curved QSH systems. Curvature does not remove TRS in curved QSH systems, since spin/charge currents with opposite polarity still propagates in opposite directions along the edges"

is misleading as it seems to imply that TRS could be violated (if spin/charge currents with opposite polarity would not propagate in opposite directions). This is not the case, of course, as the chosen model preserves TRS by construction. A simple fix is to remove the word "since" from the above sentence.

The sentence

"[...] a conventional (flat) QSH insulator conducts only CC but not SC because only the S_z component is present [...]"

is very misleading. The flat QSH system does not carry SC because L/R spins at opposite edges (see Fig. 2 a) are pointing in the opposite directions [contrary to case (c) for example], and not because only a given component is present. As a matter of fact, in panel (c) there is only one component (S_y) but there is a spin current (a fully polarized spin current).

Equation 3.

For consistency $T(E)$ should depend on the subscript α . Also, the definition of G^α [in terms of $T(E)$] is missing too.

Equation 5.

The authors should provide the detailed derivation of $g_{L,R}^{S_\alpha} = \dots$ in the Suppl. Info., which I guess they obtained from the standard eigenstate (Lehman) representation of the Berry curvature by transforming the spin operator S_z appearing in the definition of spin current.

I believe that the important relation between $\phi(L)$ and ϕ_e should be

$$\phi(L) = -\phi(R) = \theta_e / 2$$

and not

$$\phi(L) = -\phi(R) = \theta_e$$

in which case the angular variables appearing in $g_{L,R}^{S_\alpha} = \dots$ should be multiplied by two - can the authors confirm this?

* Section "Tunable SC and SHC in curved QSH systems"

The sentence "the quantization of the spin Hall conductance in a QSH system is only guaranteed when S_z is conserved" is confusing. In earlier discussion, they argue that the spin Chern number is still well defined (upon transforming operators to the flat system), which would imply a well-defined (quantized) spin Hall conductivity - at least for the particular case $\theta_e = 180$ degrees; in other cases, I believe the spin Hall current acquires both y and z polarisations. It would be interesting to check if the spin Hall conductivity tensor obeys a special relation given the existence of protected edge states. My naive guess would be (in appropriate units):

$$\sigma_{xy}^z = \cos \theta_e / 2$$

$$\sigma_{xy}^y = \sin \theta_e / 2$$

which would imply $\sigma_{xy}_{rotated\ system} = C_s$.

* Supplemental Information

It would be highly beneficial to comment on the features (plateaus, etc.) displayed by the conductance in Fig. S1; these can be easily related to the band structure (Fig. 1 main text) and would help the reader less familiar with quantum transport to grasp the significance of their results. In particular, it would be important to explain why two narrow plateaus that emerge in the interval $[-1,1]$ eV upon increase of bending angle. Finally, can the authors use dashed lines to improve readability of Fig. S1? English issues.

** Abstract

The sentence

"we discover that bending strain can be used to mitigate the spin conservation of a QSH system [...]" can be misleading since there still exists a local spin conservation law, in the sense explained below Eq. (4). I would suggest something along the lines: "we discover that bending strain can be used to control the spin orientation of counter-propagating edge states in a QSH system [...]"

The following sentence needs to be revised

"its edge states topologically protected to transport robust SC without back scattering"

** Introduction

The word "novel" in

"This novel idea can further be applied to control the magnitude of SHC in a QSH system by control of its bending curvature, which has not been achieved in a QSH system before."

is redundant and should be removed. I take this opportunity to suggest the authors to state "[...] which may pave the way for the realization and study of [...]" instead of "[...]which may open a novel route towards the realization of [...]".

Reviewer #2 (Remarks to the Author):

Review Report on: "Bending Strain Engineering of Spin Transport..." for Nature Communication by B. Huang et al.

The manuscript theoretically proposes the use of bending strain engineering in systems exhibiting the Quantum Spin Hall (QSH) effect. The authors first demonstrate the idea using as a model system a p_x, y honeycomb system in the presence of a Kane and Mele type Spin Orbit Coupling (SOC). Using tight binding calculations, the authors show that the spins of the helical edge states rotates around the S_x axes upon bending, giving rise to a non-zero two terminal conductance (as opposed to the zero value of the flat system). This first part constitutes the core of the paper, while the rest is dedicated to show how the above concept could be realized in Bi/Cl/Si(111) nano films. In particular, the authors have performed first principle calculations to determine the band structure of these nano films upon bending.

Although I think that the main idea is quite interesting, I found the paper difficult to read. In particular, I found the presentation of some of the key physical ideas sometimes confusing and self contradictory. With respect to this, I have the following comments/questions:

1 -- The authors seem to confuse spin conservation in the bulk with the fact that the QSH system does not exhibit a two terminal conductance. This statement is repeated many times throughout the paper. For example, in the introduction the authors write

“However, spin conservation mandates that there is no net SC in a QSH system”

But this is not quite right, as there is still a four terminal QSH conductance in the flat system, despite spin along the S_z direction being conserved in the bulk. I am afraid that a reader not familiar with the physics of the QSH effect could get confused by such statements and have the impression that non-zero, quantised spin currents can only exist in the setup proposed by the authors.

The four terminal measurement is explained only later on in the paper, where one can understand what the authors mean: SC is a longitudinal current measured in a two terminal setup, while SHC is the one measured in the four terminal one.

In my opinion, the authors should clearly state since the beginning that they refer to zero longitudinal spin currents in a two terminal setup.

2 -- By bending the system, and therefore rotating the spin quantization axes, it is indeed possible to obtain a non vanishing, two terminal spin conductance. This is because the S_y component rotates in opposite directions along the two edges, as correctly pointed out by the authors.

However, the spin S_z component is still conserved in the bulk, as the author also discuss after Eq.4. This seems to contradict the statement discussed in point (1) above and in several other places.

3 -- The authors write:

“In terms of robustness against back scattering, the curved and flat QSH devices are the same, as they are protected by TRS”

This claim is based on the incorrect statement (also reported in the introduction), that backscattering is forbidden in helical edge states due to time reversal invariance (TRS). The correct statement is that elastic backscattering from non magnetic impurities is forbidden. However, backscattering off non magnetic impurities is allowed in inelastic processes, i.e. in the presence of electron-electron interactions. Many body interactions are ubiquitous to one dimensional systems and therefore cannot be neglected. It was shown (see e.g. N. Kainaris et al. PRB 90, 075118 (2014) and references therein) that even in the presence of weak electron correlations, inelastic backscattering processes lead to a finite conductivity for the helical edge states.

The inclusion of these effects for the S_y polarization of the curved system may substantially change the conclusions reported in the manuscript and potentially invalidate the curved design proposed by the authors. I believe that this is an important point the authors should address in order to show that their proposal will lead to an effective advancement in the field.

4 -- When defining the model system (beginning of section II), the authors say

“The second Rashba term, associated with the existence of an electric field perpendicular to the plane, is also expected not to change significantly upon bending”.

however, no evidence, physical argument or reference is provided to support this claim. Naively, I would expect an enhancement of the rashba coupling due to curvature, just like in the graphene case (see e.g. H. Ochoa et al., PRB 86, 245411 (2012) or B. Berche et al., arXiv:1701.00363). However, I do understand that the system studied by the authors is different from graphene (whose physics is dominated by p_z orbitals) and therefore different conclusions can be reached.

However, I still believe that this is an important point to address as it could change the author’s conclusions.

5 -- On a minor note, the authors take t_{SO} in Eq. (2) to be a constant in their model. But this parameter is also likely to be affected by curvature effects. Can the authors comment on this issue?

In conclusion, although I found the main idea of the paper quite nice, I believe the authors did not provide enough evidence for this idea to be feasible (see points above) or to be significant enough to move forward the field. In particular, I am concerned about the problems connected to point (3) above. In any realistic device, the interaction effects discussed in point (3) will play a major role. In their presence, it is not clear what would be the advantages of realising the curved structures proposed by the authors, as the conclusions reported in their work may drastically change.

For these reasons, I do not recommend the manuscript for publication in Nature Communications.

Reviewer #3 (Remarks to the Author):

This is an interesting suggestion in the flourishing field of topological matter. The paper is definitely worth to be published; however, I am not completely sure that the degrees of importance and novelty are sufficient to justify the publication in Nature Communications. "Strain engineering" is the well-established concept, not only for graphene (Ref.12) but also for other 2D materials (Amorim et al, Phys. Rep. 617, 1 (2016)). Calculations are quite standard. Also, the presentation is not optimal. For instance, I have found a bit strange the statement in the introduction "An ideal ferromagnetic material is half-metal". This definitely depends on the way how you are going to use the ferromagnetic material. This is a minor point which is not relevant for the main aim of this paper but it is typical: the authors are not focused enough in their presentation. They give some obviously irrelevant references such as Ref.3 but ignore a huge massive of works on pseudomagnetic fields in 2D matter (except Refs.12,13) which makes difficult to see what are the exact novel points in their work.

Reply to Reviewer A

Summary

Question #1: “In this work, Huang & co-workers present a strain-engineering approach to spin current manipulation in quantum spin Hall (QSH) insulators. The proposed adiabatic manipulation of counter-propagating spins along the edge of a curved QSH – their main result, nicely captured in Fig. 2 (a) – is novel to my best knowledge, and is also very interesting from a theoretical standpoint.

There are a few concerns to keep in mind, notwithstanding: English language issues, mathematical imprecisions, and poor justification for neglecting local curvature effects in their minimal model [Eqs. (1)-(2)]; see below for details. However, I am confident that these issues can be overcome in a revised version, in which case I would be happy to recommend their work for publication in the prestigious Nature Communications venue.”

Reply #1: We thank the reviewer for his/her very careful review and positive recommendation. We have carefully considered all his/her comments and made revisions accordingly. The following is our responses to his/her specific comments and suggestions.

I. Introduction

Question #2: “The authors seem to use “spin current (SC)” with two different meanings. For example, in the first paragraph (Introduction) one can read: “a QSH insulator [...] transport robust SC without back scattering” and then “spin conservation mandates that there is no net SC in a QSH system” which sounds like a contradiction in terms. The authors need to clarify the distinction between pure spin currents (carrying no charge flow, e.g., as generated by spin Hall effect) and fully/partially-polarized spin currents (carrying carry charge flow, e.g., as generated in a half-metal). A simple fix would be referring to the former as pure spin currents (PSC) and to the latter as spin-polarized currents or simply “spin currents” (SC). Finally, I note that “spin Hall current” usually means a pure spin transverse (Hall) current generated by spin Hall effect, which in the context of quantum spin Hall effect can be misleading since the spin Hall conductivity (in the bulk) is vanishing.”

Reply #2: We thank the Referee for pointing out the potential confusion caused by the term of spin current (SC). Following his/her suggestion, we now use PSC to refer to pure spin currents that decouple with charge currents and SC to refer to spin currents that couple with charge currents. We also use “transverse (pure) spin current of QSH effect” to replace “spin Hall current”.

II. Results

Section “Model of curved QSH systems”

Question #3: “The text is currently somewhat confusing, requiring some improvements and possibly extra references. For instance, the authors state that term H_0 defines the band structure. This is incorrect: the band structure is defined by all the terms in the Hamiltonian. They also say that bending induces small changes to H_0 , particularly to hopping parameters: can they explain why? Wouldn't local curvature drastically affect the hopping terms, particularly those related to spin-orbit coupling? (Possible useful is the comprehensive work exploring curvature effects in graphene by Huertas-Hernando et al, PRB 74, 155426 2006.) The situation with the px-py-orbital graphene model explored by the authors is different from real pz-type graphene, but I would imagine that that bending would still couple σ (px-py-derived) and π (p_z-derived) bands, boosting Rashba-type SOC, as in pz-derived graphene. In fact, some of the approximations in Ref. 21 (where the minimal model employed by the authors is derived) could become inaccurate upon bending [specifically, I am thinking about the spin flip terms neglected when deriving Eq. (7) from (6) in Ref. 21]. In a revised version, the authors would need to clarify these points so as to fully justify their choice of model, and to provide enough context to the broad readership of Nature.”

Reply #3: We agree with the Referee that it is inaccurate to say that the H_0 term defines the band structure and we have revised this part of discussions in the description of Hamiltonian and related parameters, as follows.

$$H_0 = \sum_{n,i} \varepsilon_{ni} c_{ni}^\dagger c_{ni} - \sum_{\langle mj,ni \rangle} t_{mj,ni} (c_{mj}^\dagger c_{ni} + h.c.) \quad (1)$$

$$H_{so} = i\lambda_{so} \sum_{\langle\langle ni,nj \rangle\rangle} c_{ni}^\dagger \boldsymbol{\sigma} \cdot (\mathbf{e}_{ni} \times \mathbf{e}_{nj}) c_{nj} \quad (2)$$

In H_0 , $c_{ni}^\dagger = (c_{ni_x}^\dagger, c_{ni_y}^\dagger)$ are electron creation operators on atom n with orbital

$i \in \{p_x, p_y\}$; ε_{ni} and $t_{mj,ni}$ are electronic on-site energy and hopping integrals,

respectively. $t_{mj,ni} = A_{mj,ni}(pp\sigma) + (\mathbf{e}_{mj} \cdot \mathbf{e}_{ni} - A_{mj,ni})(pp\pi)$, where

$A_{mj,ni} = (\mathbf{e}_{mj} \cdot \mathbf{e}_{mn})(\mathbf{e}_{ni} \cdot \mathbf{e}_{mn})$. \mathbf{e}_{ni} represents the unit vector along the orbital i of atom n ,

and \mathbf{e}_{mn} is the unit vector directed from site m to n . $(pp\sigma)$ and $(pp\pi)$ are Slater-Koster integrals. In our calculations, the values of ε_{ni} are set to zero and only the nearest neighbor (NN) hoppings are taken into account. In H_{so} , λ_{so} is a constant defining the

SOC strength; $\boldsymbol{\sigma}$ is the Pauli vector. In a flat system, $\theta_e=0^\circ$, $\mathbf{e}_{ni} \times \mathbf{e}_{nj} = \pm \mathbf{e}_z$, and spins

lie strictly along the z direction. When $\theta_e \neq 0^\circ$, upon bending, all the physical observables, e.g., $p_{x,y}$ orbitals, \mathbf{e}_{ni} and spin directions $\boldsymbol{\sigma}$ are rotated accordingly by an angle of ϕ

relative to x -axis (see Fig. S1 in the Supplementary Information for the definition of ϕ).

One can see that $t_{mj,ni}$ and H_{so} are also bending dependent. On the other hand, in our model, we did not consider the p_z orbital. (Crystal field splitting may separate p_z from p_x and p_y , or absorbed atoms may remove p_z orbital, as shown for the case of Bi/Cl/Si(111) in Figure 5 in the MS.) Therefore, there is no coupling term between p_z orbital and p_x+p_y orbitals. However, we agree with the Referee that it is also interesting to consider the Rashba effect during bending. Here, we have considered a simplified NN hopping Rashba spin-orbital term in which the differences of symmetry between p_x and p_y orbitals are neglected [see Y. A. Bychkov and E. I. Rashba, J. Phys. C **17**, 6039 (1984); C. L. Kane and E. J. Mele, Phys. Rev. Lett. **95**, 226801 (2005)]:

$$H_R = i\lambda_R \sum_{\langle mj,ni \rangle} c_{mj}^\dagger (\boldsymbol{\sigma} \times \mathbf{e}_{nm})_z c_{ni} \quad (3)$$

The strength of H_R is determined by λ_R . Thus, the new Hamiltonian becomes $H=H_0+H_{so}+H_R$. Taking a curved QSH nanoribbon (Fig. 1a in MS, 40 atoms/unitcell) with a bending angle $\theta_e = 60^\circ$ as a general example, we have systemically calculated the electronic and transport properties of a bent ribbon as a function of λ_R , as shown in Figs. R1 and R2.

Fig. R1 The calculated electronic band structures of zigzag-edge QSH nanoribbons (40 atoms/unitcell, $\theta_e=60^\circ$) as a function of λ_R .

As shown in Fig. R1, when λ_R is increased from 0 to 0.9 eV (the value of λ_{so} set in our MS), the Rashba effect can significantly reduce the original SOC band gap, but the topological properties of the whole system are unchanged as long as the SOC band gap is not closed. Importantly, these calculations show that the Rashba effect will not change the spin orientation of edge states.

Also as shown in Fig. R1, when λ_R is increased, the valence bands of the QSH nanoribbons split and shift upward. when $\lambda_R > 0.6$ eV, the Rashba effect can even push the bulk valence band merging into the Fermi level, which will in turn influence the

plateau of charge conductance G and spin conductance G^{sy} around the Fermi level, as shown in Fig. R2. But it does not affect our conclusion that curvature can be applied to generate a significant non-zero spin conductance in a QSH system under a two-terminal device setting. Also, it should be noted that in real materials the Rashba effect is usually much smaller than 0.6 eV, as shown by the DFT calculated band structures in Fig. 5a in the MS. Thus, our main conclusion should not be affected significantly by the Rashba effect.

Fig. R2 The calculated charge (G) and spin (G^{sy}) conductances of zigzag-edge QSH nanoribbons ($\theta_e=60^\circ$) as a function of λ_R .

In summary, in addition to deleting some confusing statements and rewriting the description of Hamiltonian, we have added the Figs. R1 and R2 (as Fig. S5 and Fig. S6) and related discussions in the Supplementary Information. We have added a new figure (Fig. S1) in the Supplementary Information to explain the definition of rotation angle ϕ . We have also added one new section (**Rashba and disorder effects**) in Pages 8-9 in the revised MS to discuss the Rashba and other disorder effects within the scope of our model (see also our Reply to Referee B).

Question #4: “Other issues with this section. The equation (2) is incorrect: creation/annihilation operators are missing. Moreover, the authors need to define what type of orbitals goes in the index “n” – that this can be inferred from the text is no excuse to abandon rigor.”

Reply #4: Thanks for the suggestion! We have made corrections in the revision, as also explained in the Reply to Question #3 above.

Question #5: “Nano-ribbon geometry details (in Fig. 1) are missing: what type of edges, what is the direction “k” in panels b) and c)? These could be inferred from the schematic a) but still would like to see subscripts and other labels correctly displayed. 40 unit cells

in what direction?”

Reply #5: Thanks! We have now added these specifications in the revised caption of Figure 1. The edge type is zigzag. The direction of “k” in the band structure is along the length direction of nanoribbon, i.e., along x direction in Fig. 1a. The 40 atoms/unitcell means that the unit cell of nanoribbon contains 40 atoms.

Question #6: “TB parameters need to be explained (as for their meaning) in main text or in the supplemental material. For example, how does $V_{pp\sigma}$ relate to their microscopic parameters in eq. 2?”

Reply #6: We thanks the Referee for spotting this unclear presentation, which has now been revised for better clarity. We first changed the notation of Slater-Koster integral $V_{pp\sigma}$ to $(pp\sigma)$ in our revised MS. Then in our model, $t_{mj,ni} = A_{mj,ni}(pp\sigma) + (\mathbf{e}_{mj} \cdot \mathbf{e}_{ni} - A_{mj,ni})(pp\pi)$, which depends on the relative position and directional cosines between m_j and n_i orbitals at a given θ_e . When θ_e is changed, the contribution (projection) of $(pp\sigma)$ and $(pp\pi)$ to $t_{mj,ni}$ changes. So $(pp\sigma)$ and $(pp\pi)$ are directly related to the H_0 (Eq. 1) rather than H_{so} (Eq. 2).

Based on the Referee’s suggestion, we have added the above explanation in the last paragraph of Page 3.

Question #7: “Finally, can the authors make the nice panel (a) in Fig. 2 larger to help visualising the spin tilting along the edges?”

Reply #7: Thanks! We have enlarged panel (a) in Fig. 2 to make it clearer.

Section “Spin transport in curved QSH systems”

Question #8: “The phrase “Spin transport in curved QSH systems. Curvature does not remove TRS in curved QSH systems, since spin/charge currents with opposite polarity still propagates in opposite directions along the edges” is misleading as it seems to imply that TRS could be violated (if spin/charge currents with opposite polarity would not propagate in opposite directions). This is not the case, of course, as the chosen model preserves TRS by construction. A simple fix is to remove the word “since” from the above sentence.”

Reply #8: Thanks! We have changed the word “since” to “and”.

Question #9: “The sentence “[...] a conventional (flat) QSH insulator conducts only

CC but not SC because only the Sz component is present [...]” is very misleading. The flat QSH system does not carry SC because L/R spins at opposite edges (see Fig. 2a) are pointing in the opposite directions [contrary to case (c) for example], and not because only a given component is present. As a matter of fact, in panel (c) there is only one component (S_y) but there is a spin current (a fully polarized spin current).”

Reply #9: Thanks for pointing out this potential confusion. What we meant is that the Sz components are always pointing in the opposite directions at opposite edges, while S_y components induced by curvature are pointing in the same direction at opposite edges. Consequently, we made the quoted sentences above, which are equivalent to what the referee has said. We have now added these explanations to clarify our meanings.

Question #10: “Equation 3. For consistency T(E) should depend on the subscript α . Also, the definition of G^{α} [in terms of T(E)] is missing too.”

Reply #10: Thanks! We have changed the T(E) to $T_{\alpha}(E)$ in Eq. (3) and added: The retarded Green’s function is $G^r(E) = [H - (E + i\eta) + \Sigma_L + \Sigma_R]^{-1}$ and the advanced one $G^a(E) = (G^r)^{\dagger}$ in the first paragraph of Page 5 in the MS.

Question #11: “Equation 5.

The authors should provide the detailed derivation of $g_{\{L,R\}}^{\{S_{\alpha}\}} = \dots$ “in the Suppl. Info., which I guess they obtained from the standard eigenstate (Lehman) representation of the Berry curvature by transforming the spin operator S_z appearing in the definition of spin current. I believe that the important relation between $\phi(L)$ and ϕ_e should be $\phi(L) = -\phi(R) = \theta_e / 2$ and not $\phi(L) = \phi(R) = \theta_e$, in which case the angular variables appearing in $g_{\{L,R\}}^{\{S_{\alpha}\}} = \dots$ should be multiplied by two - can the authors confirm this?”

Reply #11: Thanks for the suggestion! We apologize for the mistake, and indeed the equation should be multiplied by a factor of two. We have corrected errors in the MS, and also we have added the following discussion in the Supplementary Information:

In the rotated framework, $\tilde{H} = R^{\dagger} H_C R$, the total rotated spin z-component $\tilde{S}_z = R^{\dagger} S_z R$ is conserved. For every spin up/down for \tilde{S}_z , it is an ideal integer quantum Hall system and we can use TKNN formula to calculate the edge spin conductance $g_{L,R}^{\tilde{S}_z} = \frac{\tilde{C}_s}{2} \cdot \frac{e}{2\pi}$. Before the rotation of the framework, i.e. in the

experimental framework, $S_y = \sin(\phi)\tilde{S}_z$, $S_z = \cos(\phi)\tilde{S}_z$, then we have

$$g^{S_y} = \sin(\phi)\frac{\tilde{C}_s}{2}\cdot\frac{e}{2\pi}, \quad g^{S_z} = \cos(\phi)\frac{\tilde{C}_s}{2}\cdot\frac{e}{2\pi}. \quad \phi(L) = -\phi(R) = \frac{\theta_e}{2}$$

is the curved angle for coordinates.

*** Section “Tunable SC and SHC in curved QSH systems”**

Question #12: “The sentence “the quantization of the spin Hall conductance in a QSH system is only guaranteed when S_z is conserved” is confusing. In earlier discussion, they argue that the spin Chern number is still well defined (upon transforming operators to the flat system), which would imply a well-defined (quantized) spin Hall conductivity - at least for the particular case $\theta_e = 180$ degrees; in other cases, I believe the spin Hall current acquires both y and z polarisations. It would be interesting to check if the spin Hall conductivity tensor obeys a special relation given the existence of protected edge states. My naive guess would be (in appropriate units):

$$\sigma_{xy}^z = \cos \theta_e/2$$

$$\sigma_{xy}^y = \sin \theta_e/2$$

which would imply $\sigma_{xy}^{\text{rotated system}} = C_s$.”

Reply #12: Sorry for the confusion, which we have corrected now. In the rotated system,

$$\sigma_{xy}^{\text{rotated system}} = \frac{\tilde{C}_s}{2}\cdot\frac{e}{2\pi}.$$

Before rotation in the experimental framework, $S_y = \sin(\phi)\tilde{S}_z$, $S_z = \cos(\phi)\tilde{S}_z$ where \tilde{S}_z is the spin-z component in the rotated system and ϕ is the rotated angle.

Our calculations show that only the spin component perpendicular to the QSH surface, i.e., S_z component, can contribute to the spin Hall conductance, which is also mentioned in Kane’s paper [C. L. Kane and E. J. Mele, Phys. Rev. Lett. **95**, 146802 (2005)].

*** Supplemental Information**

Question #13: “It would be highly beneficial to comment on the features (plateaus, etc.) displayed by the conductance in Fig. S1; these can be easily related to the band structure (Fig. 1 main text) and would help the reader less familiar with quantum transport to grasp the significance of their results. In particular, it would be important to explain why two narrow plateaus that emerge in the interval $[-1,1]$ eV upon increase of bending angle. Finally, can the authors use dashed lines to improve readability of Fig. S1?”

Reply #13: Thanks for the suggestion! We have added more band structures with different bending angles, Fig. R3, in the Supplementary Information as Fig. S2, which can help to better explain the results of transport calculations.

Generally, the value of charge conductance at a given energy is related to the number of bands crossing that energy. When the system is bent, there appears a small "tip" in the band dispersion around -0.5 eV and 0.5 eV, which make the number of bands crossing the energy double, as shown in Fig. R3. This additional band crossing gives rise to the increased charge conductance around -0.5 or 0.5 eV, as shown in Fig. R4.

Fig. R3 The calculated band structures of zigzag-edge QSH nanoribbons (40 atoms/unitcell) as a function of bending angle θ_e .

Fig. R4 The calculated charge conductance G of zigzag-edge QSH nanoribbons as

a function of bending angle θ_e .

Based on the Referee's comment, we have added Fig. R3 and the above discussions in the Supporting Information as Fig. S2. Also, we have changed the charge conductance (now as Fig. S3 in Supplementary Information) to the dashed line (in terms of the Referee's suggestion), as shown in Fig. R4.

English issues

****abstract**

Question #14: “The sentence “we discover that bending strain can be used to mitigate the spin conservation of a QSH system [...]” can be misleading since there still exists a local spin conservation law, in the sense explained below Eq. (4). I would suggest something along the lines: “we discover that bending strain can be used to control the spin orientation of counter-propagating edge states in a QSH system [...]””

Reply #14: Thanks for the suggestion and we have changed sentence accordingly in the abstract.

Question #15: “The following sentence needs to be revised “its edge states topologically protected to transport robust SC without back scattering””

Reply #15: We have changed this sentence to “the edge states are topologically protected, to transport robust SC without elastic back-scattering from nonmagnetic impurities”.

****Introduction**

Question #16: “The word “novel” in “This novel idea can further be applied to control the magnitude of SHC in a QSH system by control of its bending curvature, which has not been achieved in a QSH system before.” is redundant and should be removed. I take this opportunity to suggest the authors to state “[...] which may pave the way for the realization and study of [...]” instead of “[...]which may open a novel route towards the realization of [...]”.”

Reply #16: Thanks! We have deleted the word “novel” and also revised sentence to “which may pave the way for the realization and study of topological nanomechanics”.

Reply to Reviewer B

We thank the reviewer for his/her careful review and constructive comments. The following are responses to his/her specific comments and suggestions.

Question #1: “Although I think that the main idea is quite interesting, I found the paper difficult to read. In particular, I found the presentation of some of the key physical ideas sometimes confusing and self contradictory.”

Reply #1: We thank the Referee for this critical comment, which was also brought up by the Reviewer A. We have seriously considered all the comments to significantly revise the MS to improve its clarity.

Question #2: “1 -- The authors seem to confuse spin conservation in the bulk with the fact that the QSH system does not exhibit a two terminal conductance. This statement is repeated many times throughout the paper. For example, in the introduction the authors write “However, spin conservation mandates that there is no net SC in a QSH system”. But this is not quite right, as there is still a four terminal QSH conductance in the flat system, despite spin along the S_z direction being conserved in the bulk. I am afraid that a reader not familiar with the physics of the QSH effect could get confused by such statements and have the impression that non-zero, quantised spin currents can only exist in the setup proposed by the authors. The four terminal measurement is explained only later on in the paper, where one can understand what the authors mean: SC is a longitudinal current measured in a two terminal setup, while SHC is the one measured in the four terminal one. In my opinion, the authors should clearly state since the beginning that they refer to zero longitudinal spin currents in a two terminal setup.”

Reply #2: We thank the Referee for this comment. We have changed our description of spin current in a clearer context of two-terminal vs. four-terminal setup in the whole revised MS. In addition, we have changed “SHC” to “transverse pure SC of QSH effect” based on the Referee A’s comment (Referee A’s Question #2).

Question #3: “2 -- By bending the system, and therefore rotating the spin quantization axes, it is indeed possible to obtain a non vanishing, two terminal spin conductance. This is because the S_y component rotates in opposite directions along the two edges, as correctly pointed out by the authors.

However, the spin S_z component is still conserved in the bulk, as the author also discuss after Eq.4. This seems to contradict the statement discussed in point (1) above and in several other places.”

Reply #3: Sorry for the confusion. Now we clarify that “the total rotated spin z-component $\tilde{S}_z = R^\dagger S_z R$ is conserved” , where \tilde{S}_z is the rotated spin in the rotated

framework and S_z is in the experimental framework. Spin \tilde{S}_z is only conserved in the rotated framework, not conserved in the experimental framework. Please also see our responses to Referee A's questions #9, 11, 12, which are related to this comment.

Question #4: “3 -- The authors write: “In terms of robustness against back scattering, the curved and flat QSH devices are the same, as they are protected by TRS”. This claim is based on the incorrect statement (also reported in the introduction), that backscattering is forbidden in helical edge states due to time reversal invariance (TRS). The correct statement is that elastic backscattering from non-magnetic impurities is forbidden. However, backscattering off non-magnetic impurities is allowed in inelastic processes, i.e. in the presence of electron-electron interactions. Many body interactions are ubiquitous to one dimensional systems and therefore cannot be neglected. It was shown (see e.g. N. Kainaris et al. PRB 90, 075118 (2014) and references therein) that even in the presence of weak electron correlations, inelastic backscattering processes lead to a finite conductivity for the helical edge states.

The inclusion of these effects for the S_y polarization of the curved system may substantially change the conclusions reported in the manuscript and potentially invalidate the curved design proposed by the authors. I believe that this is an important point the authors should address in order to show that their proposal will lead to an effective advancement in the field.”

Reply #4: We thank the referee for this stimulating comment. We agree with the Referee that inelastic back scattering can still occur in the presence of many-body interactions in a QSH system, which may induce a finite conductivity as discussed in N. Kainaris et al (2014). Accordingly, we have revised the related statements in the paper and noted the possible difference from inelastic scattering along with the cited reference. We have now added substantial new calculation results to address the diverse effects of disorder and show our conclusion remain robust (see detailed discussion below).

On the other hand, however, we believe that to fully address the effects of inelastic scattering due to many-body interactions is beyond the scope of our present work. The experimental measurements in InAs/GaSb bilayer QSH system indicate that the quantized conductance plateau can survive in a broad regime even the helical edge modes of this system are in a strongly interacting regime [L. Du et al, Phys. Rev. Lett. **114**, 096802 (2015)]. Moreover, we would also like to point out that the many-body effects in our considered system should be relatively weak, because what considered here is p -orbital QSH systems (both model calculations and Bi/Cl/Si(111) first-principles calculations), which is usually not a strongly correlated system, so that it will have typically weaker many-body effects than those containing d -orbital or f -orbital electrons. Therefore, we do not expect the many-body effects to significantly change

the main conclusion of our MS, i.e., curvature can be applied to turn the spin current and edge spin current of a QSH nanoribbon based on p -orbital materials.

We also agree that many-body interactions are ubiquitous to one dimensional systems, however, we think the problem we consider here is not exactly one dimensional, since topological edge states are actually bulk Bloch wavefunctions terminated at the edge subject to the well-known bulk-boundary correspondence.

Nevertheless, we feel the referee raised a very good point and we have now added a new section (**Rashba and disorder effects**) in Pages 8-9 to address the referee's question to the best possibility within the context of our study, by further considering the Rashba effect, random-on-site-energy effect and random-atomic-displacement effect as detailed below.

Firstly, we now show that a reasonably strong Rashba spin-orbital effect will not change our conclusion, as explained in our responses to Referee A's Comment #3 (also see Figs. S5 and S6 in the Supplementary Information).

Fig. R5 The calculated band structures of zigzag-edge QSH nanoribbons (40 atoms/unitcell, $\theta_e=60^\circ$) as a function of random on-site energy ϵ_n .

Secondly, we have considered the effects due to random on-site energies, i.e., by randomly changing the on-site energies, ϵ_{ni} of all the atoms in $H_0 = \sum_{n,i} \epsilon_{ni} c_{ni}^\dagger c_{ni} - \sum_{\langle mj,ni \rangle} t_{mj,ni} (c_{mj}^\dagger c_{ni} + h.c.)$. We considered two cases by varying the onsite energies from the initial values up to $0.25 \lambda_{so}$ or $1.0 \lambda_{so}$ (λ_{so} is the SOC strength). Usually, the changes of on-site energies may be induced by the effects from substrate

or impurity or by many-body interactions through self-energy corrections. Here we have randomly selected two configurations to calculate their electronic and transport properties. As an example, Figure R5 shows the calculated band structures for a bent QSH ribbon with bending angle $\theta_e = 60^\circ$. The random on-site energies will effectively lift the degeneracy of edge states, but it will not affect the spin rotations of edge states (the main conclusion of our MS).

Fig. R6 The calculated charge (G) and spin (G^{sy}) conductances of zigzag-edge QSH nanoribbons ($\theta_e=60^\circ$) as a function of different random ϵ_n .

We also performed transport calculations, as shown in Fig. R6. The random on-site energies will make some quantitative change in the charge conductance G and spin conductance G^{sy} curves, but they will not change the overall shape of conductance spectrums or alter our main conclusions, i.e., *curvature can be applied to generate a significant non-zero spin conductance in a QSH system under a two-terminal device setting.*

Finally, we have considered the random-atomic-displacement effect, which can account for the electron-phonon interactions or thermal effect. We assume all the atoms are displaced from their equilibrium positions in any given direction by a maximum distance of 0.125 \AA , i.e., $R_p=0.125 \text{ \AA}$. This will effectively change the hopping term, $t_{mj,ni}$, even if the changes in $(pp\pi)$ and $(pp\sigma)$ are negligible.

Fig. R7 The calculated electronic band structures of zigzag-edge QSH nanoribbons (40 atoms/unitcell, $\theta_e=60^\circ$) without and with random atomic position effect.

Similar to the random-on-site-energy effect, the random-atomic-displacement effect also lifts the degeneracy of edge states, as shown in Fig. R7. Again, we have randomly selected one configuration to calculate its electronic and transport properties. However, it will not affect the transport properties of QSH nanoribbons, as shown in Fig. R8.

Fig. R8 The calculated charge (G) and spin (G^{sy}) conductances of zigzag-edge QSH nanoribbons ($\theta_e=60^\circ$) without and with random atomic position effect.

In summary, based on the Referee's comment, we have added one additional section (**Rashba and disorder effects**) in Pages 8-9 to discuss the Rashba, disorder effects, and electron-electron interaction effects. We have cited N. Kainaris et al. PRB **90**, 075118 (2014) as Ref. 33 and L. Du et al, Phys. Rev. Lett. **114**, 096802 (2015) as Ref. 34 in the revision. We have added Figs. R5-R8 in the Supplementary Information as Figs. S7-Figs. S10. We have changed all the description ".....without back scattering....." to ".....without elastic back-scattering from non-magnetic impurities....." in the revision.

Question #5: "4 -- When defining the model system (beginning of section II), the authors say

"The second Rashba term, associated with the existence of an electric field perpendicular to the plane, is also expected not to change significantly upon bending". however, no evidence, physical argument or reference is provided to support this claim. Naively, I would expect an enhancement of the rashba coupling due to curvature, just like in the graphene case (see e.g. H. Ochoa et al., PRB 86, 245411 (2012) or B. Berche et al., arXiv:1701.00363). However, I do understand that the system studied by the authors is different from graphene (whose physics is dominated by p_z orbitals) and therefore different conclusions can be reached.

However, I still believe that this is an important point to address as it could change the author's conclusions."

Reply #5: Thanks for bringing up this point. We have now added additional calculation results and discussions to address the possible effects of curvature on Rashba terms, as we explained in our responses to Referee A's Question #3 (also in Figs. S5 and S6 in the Supplementary Information). Our conclusion is Rashba effect will not affect our conclusion.

Question #6: "-- On a minor note, the authors take t_{so} in Eq. (2) to be a constant in their model. But this parameter is also likely to be affected by curvature effects. Can the authors comment on this issue?"

Reply #6: t_{so} is a materials' (or atomic) property, and usually the heavier the atom is, the larger the t_{so} . So, we don't expect it to change much upon bending. On the other hand, we have shown in our response to the last question that SOC effect will not affect our main conclusions. Thus, some variations of t_{so} would not affect our conclusions either.

Question #7: "In conclusion, although I found the main idea of the paper quite nice, I believe the authors did not provide enough evidence for this idea to be feasible (see points above) or to be significant enough to move forward the field. In particular, I am concerned about the problems connected to point (3) above. In any realistic device, the

interaction effects discussed in point (3) will play a major role. In their presence, it is not clear what would be the advantages of realising the curved structures proposed by the authors, as the conclusions reported in their work may drastically change.”

Reply #7: We thank the referee for his critical review of our MS, and we believe we have now addressed all his questions, especially the point (3) above to mitigate his concerns. To reiterate, we believe to fully address the many-body interactions is beyond the scope of our present work. However, we have added new calculation results to confirm the feasibility of our idea (Thanks to the referee for considering our idea to be novel and quite nice!). Specifically, we have shown our key conclusions are robust against variations in Rashba interactions and diverse disorder effects to partially account for many-body and other interaction effects.

Based on the Referees’ comments and suggestions, we have done our best to make significant revisions to our MS. We have one additional section (**Rashba and disorder effects**) in Pages 8-9 in the MS and also add 9 new figures and discussions in the Supplementary Information to support and enhance our conclusion. We wish that the Referee now can be satisfied with our effort.

Reply to Reviewer C

Question #1: “This is an interesting suggestion in the flourishing field of topological matter. The paper is definitely worth to be published; however, I am not completely sure that the degrees of importance and novelty are sufficient to justify the publication in Nature Communications. "Strain engineering" is the well-established concept, not only for graphene (Ref.12) but also for other 2D materials (Amorim et al, Phys. Rep. 617, 1 (2016)). Calculations are quite standard. Also, the presentation is not optimal. For instance, I have found a bit strange the statement in the introduction "An ideal ferromagnetic material is half-metal". This definitely depends on the way how you are going to use the ferromagnetic material. This is a minor point which is not relevant for the main aim of this paper but it is typical: the authors are not focused enough in their presentation. They give some obviously irrelevant references such as Ref.3 but ignore a huge massive of works on pseudomagnetic fields in 2D matter (except Refs.12,13) which makes difficult to see what are the exact novel points in their work.”

Reply #1: We thank the Referee for reviewing our MS and consider it interesting. We agree with the referee that strain engineering is a very versatile approach and has been applied to many different materials including different 2D materials. However, possibly due to our not completely clear presentation, it seemed to us the referee might have misunderstood that our proposed approach is somewhat related to “strain induced pseudomagnetic fields” effect. This might be the main reason that he/she is “not completely sure that the degrees of importance and novelty are sufficient to justify the publication in Nature Communications.” We would like to further clarify that the

approach of strain engineering we propose here is novel, and to the best of our knowledge has not been shown before. This point of view has also being attested by both Referee A and B. Specifically, it is unrelated to “strain induced pseudomagnetic fields” effect. This is the reason that in the introduction, we cited only two papers on this topic along with other citations of previous works of strain engineering, all of which are generally relevant but none directly related to our new approach. Considering the referee’s suggestion, we have now added a couple of more references to this specific topic and the one reference the referee suggested. We have also revised introduction and deleted some irrelevant discussions. We have added R. Roldan et al, JPCM 27, 313201 (2015), B. Amorim et al, Phys. Rep. 617, 1 (2016), M.A.H. Vozmediano et al, Phys. Rep. 496, 109 (2010), F. Guinea et al, PRB 81, 035408 (2010) as Refs. 10, 11, 12, 13, respectively, in the revised MS.

Below, to better address the referee’s question and mitigate his/her concerns on the novelty of our work, we summarize several novel points of our approach beyond all the previous studies of strain engineering:

- (1) We have demonstrated that bending strain can be applied to tune spin orientations of topological edge states in a QSH ribbon, which in turn can be used to tune edge spin currents (coupled with charge currents in a two-terminal device setting, or decoupled with charge currents in a four-terminal device setting) in a QSH device. The tunable edge spin currents by strain has never been found before. These are novel findings in terms of both the strain induced physical phenomena (modifying *topological edge-state* and *spin transport* properties vs. modifying *electronic* and *electronic transport* properties) and the form of strain (*bending* strain vs. *in-plane tensional* strain) being applied.
- (2) We further proposed a novel and effective approach to grow curvature-controllable QSH nanoribbons by nanomechanical architecture process. Taking a real material of Bi QSH monolayer deposited on Cl/Si(111) surface as a real material example, we have systemically demonstrated the feasibility of our proposal by first-principles structural and electronic structural calculations. Our results demonstrate that we can grow curvature-controllable QSH nanoribbons in the real materials to realize the novel spin properties of curved QSH systems as we proposed.

Thus, our study not only significantly advances our fundamental knowledge of spin transport properties in QSH systems, but also broadens the scope of nanotechnology in topological materials and devices and vice versa.

Also, we have done our best to make significant revisions (also see our Replies to Referees A and B) to our MS, including add one new section for **Rashba and disorder effects** in Pages 8-9 and added 9 new figures in the Supplementary Information to enhance our conclusion. We wish that the Referee now will be satisfied with our revision.

SUMMARY

1. We have replied all the comments from the Referees and made significant revisions in our manuscript.
2. We have added a new section, **Rashba and disorder effects**, in the manuscript (Pages 8-9) to response to the Referees' comments.
3. We have added **9 new figures** and related discussions in the Supplementary Information to enhance our conclusion.
4. We have updated the **Figure 2** based on the Referee A's suggestion.
5. We have deleted two irrelevant references and added **6 new references (Refs. 10-13, 33, 34)** based on the Referees B and C's comments.

REVIEWERS' COMMENTS:

Reviewer #1 (Remarks to the Author):

The authors have replied satisfactorily to my questions. I recommend the present version of the manuscript for publication in Nature Communications.

Reviewer #2 (Remarks to the Author):

I read with pleasure the response to the referee report. I found that the authors did answer most of my and the other's referees questions in a satisfactory way. In particular, I have really appreciated the extra calculations performed in order to partially address the issue of many body effects and random Rashba disorder. I understand that this is a difficult issue whose full understanding is beyond the scope of this paper.

As a remark, in their response the authors write:

"we think the problem we consider here is not exactly one dimensional, since topological edge states are actually bulk Bloch wavefunctions terminated at the edge subject to the well-known bulk-boundary correspondence".

I do not agree with such statement, as edge states are truly one dimensional and present strongly correlated features as it is well known in the case of the Quantum Hall effect (QHE). This holds true not only for the fractional QHE but also for some integer filling fractions supporting multiple edge modes.

The one dimensional nature of the states is due to the fact that bulk states are gapped (incompressible); therefore a gapless state on the edge cannot diffuse back in the bulk. This confinement guarantees the one dimensional nature of the edge states and therefore the enhanced electron-electron correlation even though the bulk itself may not be in a strongly correlated state.

Nevertheless, as I have stated above, this is a difficult issue that is well beyond the scope of the paper.

In its current revised form I would like to recommend the paper for publication in Nature communications.

Reviewer #3 (Remarks to the Author):

I think the manuscript is essentially improved. I have found the author's reply to the referee critical remarks (including my own) quite convincing. In my opinion, the present version can be recommended for the publication.

Reply to Reviewer B

Question #1: “I read with pleasure the response to the referee report. I found that the authors did answer most of my and the other's referees questions in a satisfactory way. In particular, I have really appreciated the extra calculations performed in order to partially address the issue of many body effects and random Rashba disorder. I understand that this is a difficult issue whose full understanding is beyond the scope of this paper.

As a remark, in their response the authors write:

“We think the problem we consider here is not exactly one dimensional, since topological edge states are actually bulk Bloch wavefunctions terminated at the edge subject to the well-known bulk-boundary correspondence”.

I do not agree with such statement, as edge states are truly one dimensional and present strongly correlated features as it is well known in the case of the Quantum Hall effect (QHE). This holds true not only for the fractional QHE but also for some integer filling fractions supporting multiple edge modes. The one dimensional nature of the states is due to the fact that bulk states are gapped (incompressible); therefore a gapless state on the edge cannot diffuse back in the bulk. This confinement guarantees the one dimensional nature of the edge states and therefore the enhanced electron-electron correlation even though the bulk itself may not be in a strongly correlated state.

Nevertheless, as I have stated above, this is a difficult issue that is well beyond the scope of the paper. In its current revised form I would like to recommend the paper for publication in Nature communications.”

Reply #1: We thank the Referee for the recommendation of accepting our manuscript in its current form. We agree with the Referee’s comment on our previous response to his question. Since this is only mentioned in the response, no change is needed in our manuscript.